# Feeding pattern and associated factors among children aged 6–23 months in the Tahtay Maichew district, northern Ethiopia

Shewit Engdashet Berhe[1]*, Teklit Grum[2], Teklehaymanot Huluf Abraha[2], Gebrekiros Aregawi[3], Ebud Ayele Dagnazgi[1], Kiros Gereziher Arefayne[1], Ermyas Brhane[1]

**1** Department of Human Nutrition, School of Public Health, College of Health Sciences, Aksum University, Aksum, Ethiopia, **2** Department of Reproductive Health, School of Public Health, College of Health Sciences, Aksum University, Aksum, Ethiopia, **3** Department of Midwifery, College of Health Sciences, Aksum University, Aksum, Ethiopia

* shewit2023@gmail.com

## Abstract

The first one thousand days of life are a critical window of opportunity for children's health and development. Nutritional deficiencies during this time can have serious consequences for the child's health and development, with limited chances for correction later. For example, inadequate feeding among children can lead to consequences such as stunting, wasting, impaired immunity, and delayed cognitive development. Therefore, this study aimed to determine meal frequency and associated factors among children aged 6–23 months in the Tahtay Maichew district, northern Ethiopia. We conducted a community-based cross-sectional study involving 981 randomly selected mothers of children aged 6–23 months. Data were collected using a structured, interviewer-administered questionnaire. The children's meal frequency was determined by asking mothers how many times their child had eaten food in the 24 hours preceding the survey. We used binary logistic regression with backward elimination to identify factors associated with children's meal frequency. Overall, 68% (95% CI: 64.9, 70.9%) of the children received adequate meal frequency. Being from a rich household (p = 0.013, 95% CI = 1.12, 2.59), having growth monitoring follow-up (p < 0.001, 95% CI = 1.44, 2.88), good mother's knowledge of child feeding (p < 0.001, 95% CI = 1.59, 3.22), and having a birth preparedness plan (p < 0.046, 95% CI = 1.013, 4.339) were associated with adequate meal frequency. The proportion of children who did not receive adequate meal frequency was significantly high. Being from a wealthy household, undergoing growth monitoring follow-up, having a knowledgeable mother regarding child feeding, and having a birth preparedness plan were associated with adequate meal frequency. In conclusion, our findings highlight the need to improve child meal frequency through enhancing maternal knowledge of child feeding, strengthening growth monitoring and promotion services, and improving socioeconomic status, as indicated by the wealth index.

**Data availability statement:** The datasets used during the current study are uploaded as Supporting information.

**Funding:** This work was supported by Aksum University to EB. This funding has no grant or project number. The funder had no role in the study design, data collection and analysis, the decision to publish, or the preparation of the manuscript.

**Competing interests:** The authors have declared that no competing interests exist.

## Background

The first one thousand days of life are a crucial window of opportunity for children's health and development. This period is critical for establishing lifelong health, during which the body, brain, metabolism, and immune system develop significantly. A child's ability to develop, learn, and thrive depends heavily on receiving proper nutrition during this critical period. Nutritional deficiencies at this stage can have serious and often irreversible consequences for the child's health and development [1–3]. For example, inadequate feeding practices among children can lead to consequences such as stunting, wasting, impaired immunity, and delayed cognitive development [4–7].

Meal frequency is a proxy indicator of the adequacy of Infant and Young Child Feeding Practices among children aged 6–23 months [8]. Improving feeding practices, including meal frequency, is therefore fundamental to enhancing child health and development outcomes [9]. However, evidence from African countries shows that only 38.6% of children in Tanzania [10], and 57.95% in Gambia [11] receive the recommended meal frequency. Other studies in Ethiopia revealed that only 50.4% of children in northwestern Ethiopia [12], 47% in Bahirdar city [13], and 45% in the whole of Ethiopia [14] received an age-appropriate meal frequency.

Infant and Young Child Feeding practices, such as exclusive breastfeeding and continued breastfeeding until the age of two years, are at rates of 58% and 76% in Ethiopia, respectively [14]. Ethiopia is committed to the International Code of Marketing of Breast-milk Substitutes and has had a legal framework supporting the code's principles since 2016. However, there are limitations on the regulation of follow-up formula and growing-up milk. On the other hand, breastfeeding is culturally encouraged in Ethiopia, and most mothers, especially in rural areas, are willing to breastfeed, even though there are concerns regarding its appropriateness [15–17].

Although adequate meal frequency in early life is crucial for children's healthy growth and development, there is a research gap regarding the determinants and prevalence of meal frequency within the specific environmental and socioeconomic context of rural Tigray, particularly the Tahtay Maichew district. There are indeed studies conducted in Ethiopia, but most were done in regions outside of Tigray, particularly in the southern and northwestern parts of the country [12,13,18–20]. In addition, while valuable, large-scale surveys like the Ethiopian Demographic and Health Survey may overlook local socioeconomic and cultural variations. Therefore, this study aimed to determine meal frequency and associated factors among children aged 6–23 months in the Tahtay Maichew district, northern Ethiopia.

## Methods and materials

### Ethics approval and consent to participate

Before the commencement of the study, ethical approval was obtained from the Institutional Review Board of the College of Health Sciences at Aksum University. Additionally, a letter of permission was received from the Tahtay Maichew district health office. Written informed consent was also obtained from the parents of the children,

after providing a thorough explanation of the study's objectives to ensure their willingness to participate. Additionally, information collected from participants was held anonymously to maintain confidentiality.

### Study setting

This study was conducted in the Tahtay Maichew district, located in the central zone of the Tigray Regional State, northern Ethiopia. The district has 17 kebeles (the smallest administrative unit in Ethiopia) administrations. The main source of income for more than 95% of the population in the area is agriculture, while maize, 'Teff', and sorghum are the staple cereals. Generally, the Tigray region has a higher poverty level because its economy mainly depends on agriculture, is drought-prone, receives limited rainfall, and farmers have small landholdings that limit their production [21,22].

### Study design and population

A community-based cross-sectional study design was employed. The study participants were mothers of children aged 6–23 months who lived in the selected kebeles of the Tahtay Maichew district, northern Ethiopia.

### Sample size and sampling technique

The sample size for determining children's meal frequency was calculated using a single population proportion formula with the following assumptions: 45% proportion of children aged 6–23 months who received meal frequency appropriate for their age [14], 95% confidence level, 4% margin of error, design effect of 1.5, and 10% nonresponse rate. Finally, the sample size was determined to be 981. The sample size for the analytical part was calculated using a double-population proportion formula, but it was smaller than the sample size for the descriptive part. Therefore, the final sample size was set at 981.

Initially, out of the 17 kebeles in the Tahtay Maichew district, 8 were chosen using a simple random sampling method. Subsequently, the total sample size was distributed proportionately among the selected kebeles based on the number of children aged 6–23 months. Following this, a rapid census was conducted on the selected kebeles 07 days before the actual data collection to identify households with children aged 6–23 months and use it as a sampling frame. Finally, child-mother pairs were selected from each kebele using a systematic random sampling technique, after assigning a code to each household with a child aged 6–23 months (**Fig 1**).

### Measurements

Data on socio-demographic and economic characteristics, reproductive factors, health service utilization, knowledge of child feeding, media exposure, and maternal social capital were collected using an Interviewer-administered structured questionnaire adapted from different studies [10,12–14,18–20,23–27]. The questionnaire was prepared in English and translated into Tigrigna (the local language). Subsequently, it was translated back to English to check its consistency.

Feeding pattern was assessed by measuring meal frequency, asking the mothers how many times the child consumed solid, semisolid, or soft foods in the 24 hours preceding the survey. Accordingly, consuming solid, semisolid, or soft foods two or more times for breastfed infants aged 6 to 8.9 months, three or more times for breastfed children aged 9 to 23.9 months, and four times for non-breastfed children aged 6 to 23.9 months was considered as receiving adequate meal frequency [28].

Wealth index data were collected on 24 household assets, including livestock, household equipment, annual cereal production, vehicle ownership, agricultural land ownership, and housing conditions. Each of these assets was recorded as either 0 (not owned) or 1 (owned). In the analysis, we assessed the suitability of the variables for Principal Component Analysis (PCA) using the Kaiser-Meyer-Olkin (KMO) and communality values. Finally, we summed these factor scores and categorized them into three groups: poor, medium, and rich [19].

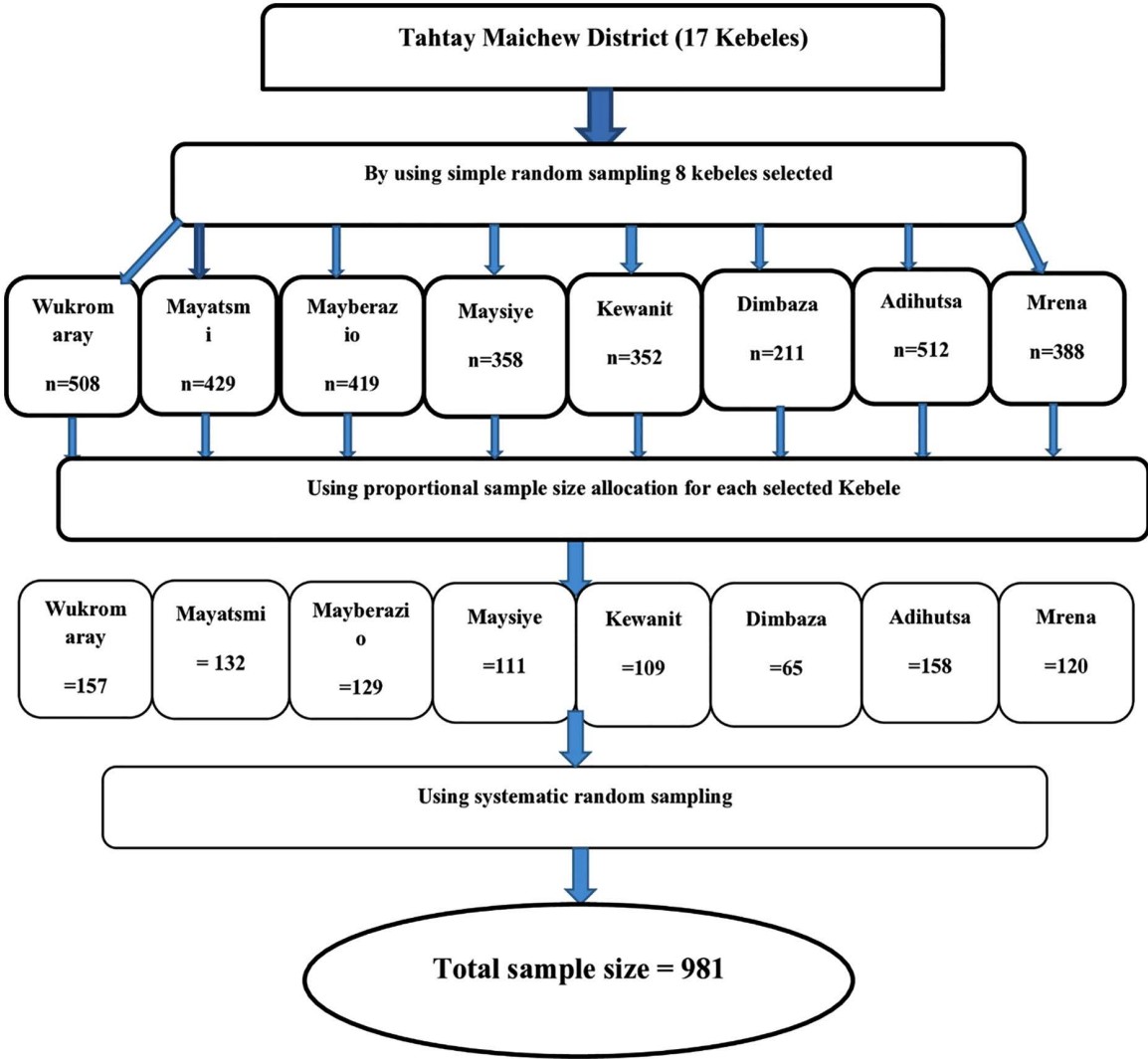

**Fig 1. Schematic representation of the sampling technique for the meal frequency of children.**

Mothers' knowledge of complementary feeding was determined by asking questions related to complementary feeding and meal frequency. A mother who scored above the mean for the knowledge-related questions was considered to have good knowledge of child feeding [29]. Additionally, maternal social capital was assessed using six questions. A mother who scored above the mean for the social capital-related questions was considered to have good social capital in the last 12 months [30]. Regarding media exposure, a mother who read a newspaper or magazine at least once a week, listened to the radio, or watched television was considered to have satisfactory media exposure [12].

## Data management and analysis

After all the questionnaires were checked for completeness and consistency, the data were coded and entered into EPI INFO version 7 and then exported to STATA-12 statistical software for analysis. In the descriptive analysis, data normality was assessed using the skewness test and P-P plots. Continuous variables were summarized using means and standard deviations, while categorical variables were presented as frequencies and percentages.

To identify factors associated with meal frequency, we first performed univariable analysis using chi-square tests or t-test as appropriate. Subsequently, we fitted a multivariable logistic regression using the backward elimination method that included all variables demonstrating significant associations in the univariable analysis. The goodness-of-fit of the multivariable model was assessed using the Hosmer-Lemeshow test. Additionally, we tested for multicollinearity using variance inflation factors (VIF < 10) and tolerance statistics (> 0.1).

## Results

A total of 949 mother-child pairs participated in the study, yielding a response rate of 96.7%. The mean [±SD] ages of the children and mothers were 13.7 ± 4.6 months and 29.7 ± 6.7 years, respectively. Moreover, the mean parity of mothers was 2.8 ± 1.7. The majority (70.5%) of children were exclusively breastfed for the first six months, and 94.5% of them were still being breastfed at the time of the survey. Orthodox Christianity was the predominant religion (92.5%) (**Table 1**).

Overall, 68% (95% CI: 64.9, 70.9%) of the children aged 6–23 months had adequate meal frequency, and only 43.4% of them had adequate dietary diversity. Most of the children (96.94%) consumed grains, tubers, and roots, while only a small percentage consumed flesh foods, fruits, and vegetables, indicating limited dietary diversity (**Fig 2**).

The health service utilization trends showed that 90% of the mothers had at least one antenatal care (ANC) visit, 83.5% delivered at a health facility, and 45.3% had one or more postnatal care (PNC) follow-up. (**Table 2**).

### Factors associated with the meal frequency of children

In the multivariable logistic regression analysis, being from a wealthy household, undergoing growth monitoring follow-up, having a knowledgeable mother regarding child feeding, and having a birth preparedness plan were significantly associated with adequate meal frequency.

Accordingly, the odds of receiving adequate meal frequency were 1.70 times higher (p = 0.013, 95% CI = 1.12, 2.59) among children belonging to rich families than children from poor families. Moreover, children who attended growth monitoring follow-ups had twice the odds (p < 0.001, 95% CI = 1.44, 2.88) of receiving adequate meal frequency compared to their counterparts. The odds of receiving adequate meal frequency were 2.26 (p < 0.001, 95% CI = 1.59, 3.22) times higher among children of mothers with good knowledge of child feeding than among children of mothers with poor knowledge. Additionally, children born to mothers who had a birth preparedness plan had twice (p < 0.046, 95% CI = 1.013, 4.339) the odds of receiving adequate meal frequency than their counterparts (**Table 3**).

## Discussion

In this study, we assessed Meal frequency and associated factors among children aged 6–23 months in the Tahtay Maichew district, Tigray, northern Ethiopia. Our results revealed that a considerable proportion of children aged 6–23 months did not have adequate meal frequency. This indicates that the children with suboptimal feeding during this critical period are at an increased risk of macronutrient undernutrition, micronutrient deficiencies, and impaired cognitive and physical development, which could lead to long-term health and economic burdens for both individuals and society [1,2,31].

The proportion of children in this study who received adequate meal frequency (68%) is comparable with findings from Wolaita-Sodo (68.9%) [20], and Bale (68.4%) [32], but lower than results from Dabat (72.2%) [19], and Addis Ababa (90.6%) [33]. Conversely, the finding was higher than the 50.4% in Dangla [12], and 45% in the 2016 EDHS [14].

On the other hand, among the children who had adequate meal frequency, only 43.4% achieved an adequate level of dietary diversity. This finding highlights an important gap; even when children are fed adequately during the day, the foods they eat may still lack variety. In other words, meeting meal frequency does not necessarily ensure a nutritionally balanced diet among children. This underscores the need for our interventions to promote not only increased feeding frequency but also diverse feeding practices.

**Global Public Health**

**PLOS**

**Table 1. Socio-demographic characteristics of participants.**

| Variable | Meal frequency | | P-value* |
|---|---|---|---|
| | Adequate, N (%)/ mean±SD | Inadequate, N (%)/ mean±SD | |
| **Age of mother (years)** | 29.5±6.5 | 30.2±6.9 | 0.141 |
| **Age of child (months)** | 13.8±4.7 | 13.4±4.4 | 0.190 |
| **Sex of child** | | | 0.227 |
| Male | 339 (35.7) | 147 (15.5) | |
| Female | 306 (32.2) | 157 (16.6) | |
| **Family size** | 4.7±1.7 | 4.8±1.8 | 0.318 |
| **Marital status of mothers** | | | 0.113 |
| Married | 579 (61.0) | 269 (28) | |
| Divorced | 46 (4.9) | 30 (3.2) | |
| Separated | 12 (1.6) | 5 (0.5) | |
| Widowed | 8 (0.8) | 0 | |
| **Mothers' educational status** | | | <0.001 |
| No formal education | 197 (20.8) | 134 (14.1) | |
| Primary school (1–8) | 218 (23) | 99 (10.4) | |
| Secondary school (9–12) | 186 (19.6) | 66 (7.0) | |
| Diploma and above | 44 (4.6) | 5 (0.5) | |
| **Husbands' educational status** | | | 0.045 |
| No formal education | 113 (13.1) | 69 (8.0) | |
| Primary school (1–8) | 230 (26.6) | 110 (12.7) | |
| Secondary school (9–12) | 187 (21.6) | 79 (9.1) | |
| Diploma and above | 61 (7.0) | 16 (1.9) | |
| **Mothers' occupation** | | | 0.002 |
| Farmer | 340 (35.8) | 186 (19.6) | |
| Housewife | 168 (17.7) | 76 (8) | |
| Self-employed | 54 (5.7) | 16 (1.7) | |
| Daily worker | 35 (3.7) | 20 (2.1) | |
| Government employed | 48 (5.1) | 6 (0.6) | |
| **Husbands' occupation** | | | 0.004 |
| Farmer | 384 (44.4) | 203 (23.5) | |
| Self-employed | 76 (8.8) | 28 (3.2) | |
| Daily worker | 64 (7.4) | 31 (3.6) | |
| Government employed | 67 (7.7) | 12 (1.4) | |
| **Wealth index** | | | <0.001 |
| Poor | 189 (19.9) | 128 (13.5) | |
| Medium | 218 (23.0) | 105 (11.0) | |
| Rich | 238 (25.1) | 71(7.5) | |
| **Mothers' knowledge of child feeding** | | | <0.001 |
| Good | 437 (46.0) | 112 (11.8) | |
| Poor | 208 (21.9) | 192 (20.3) | |
| **Media exposure** | | | <0.001 |
| Yes | 241 (25.4) | 66 (7.0) | |
| No | 404 (42.5) | 238 (25.1) | |
| **Maternal social capital** | | | <0.001 |
| Good | 361 (38.0) | 107 (11.3) | |
| Poor | 284 (29.9) | 197 (20.8) | |

* Chi-square or t-test.

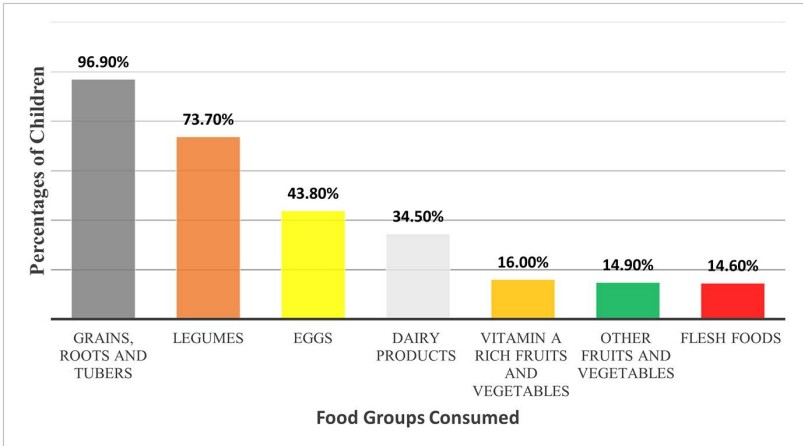

**Fig 2. Food groups consumed by children aged 6-23 months, Tahtay Maichew District, northern Ethiopia.**

The association of factors such as being from a wealthy household, undergoing growth monitoring follow-up, having a knowledgeable mother regarding child feeding, and having a birth preparedness plan highlights the interplay between economic, educational, and health system factors. Moreover, these identified predictors present challenges and opportunities for various public health interventions.

The association between higher household wealth and adequate meal frequency indicates that economic constraints may limit households' access to food and affect child feeding. it also highlights the need to strengthen social protection programs such as the Productive Safety Net Program, promoting income-generating activities for poorer families, and generally focusing on poverty alleviation in the area, as these interventions are effective in other studies from low-income settings [34–37]. In Tigray, most households do not meet the daily recommended calorie intake [21]. Tigray experiences drought, characterized by limited, unreliable, and variable rainfall, resulting in repeated crop failures and asset depletion [38]. The shortage of grazing land and water reduces livestock productivity, which ultimately affects per-capita food consumption and food security [39]. This association between wealth index and meal frequency is also consistent with findings reported in West Africa [26], Tanzania [10], Gambia [11], and northwestern Ethiopia [19].

Growth monitoring follow-up was also positively associated with receiving adequate meal frequency. This may be because growth monitoring and promotion services provide opportunities for health workers to detect growth faltering early and offer individualized counseling on appropriate feeding practices, thereby enhancing the mothers' knowledge on how to feed their children [40]. Regular growth monitoring and promotion (GMP) is carried out in the study area by health extension workers who perform GMP for children under two years old every month and use the family health card to track all children. It is also conducted at health centers incorporated with the integrated management of newborn and childhood illness (IMNCI), despite limitations caused by a lack of equipment, inaccurate measurements, and low service utilization. Supportive findings were also reported from studies in northwest Ethiopia [19] and Nepal [41].

Mothers' good knowledge of child feeding was also a key factor influencing whether their children received adequate meal frequency. This suggests a need for public health interventions, such as nutrition education and behavioral change communication, along with improving women's literacy to enhance their understanding of feeding practices [42,43]. Such efforts can be delivered through various contact points, including antenatal and postnatal care, delivery, and community health days. Having a birth preparedness plan was also associated with adequate meal frequency, which may be because individuals who plan for their birth are also more likely to have appropriate birth spacing and a smaller family size.

**Table 2. Reproductive and health Service Utilization related characteristics of participants.**

| Variable | Meal frequency | | p-value* |
|---|---|---|---|
| | Adequate, N (%) | Inadequate, N (%) | |
| **ANC follow-up** | | | <0.001 |
| Yes | 606 (63.9) | 250 (26.3) | |
| No | 39 (4.1) | 54 (5.7) | |
| **Number of ANC follow-ups (n = 856)** | | | 0.017 |
| <4 | 208 (24.3) | 114 (13.3) | |
| ≥4 | 398 (46.5) | 136 (15.9) | |
| **Birth preparedness plan** | | | <0.001 |
| Yes | 590 (62.2) | 231 (24.3) | |
| No | 55 (5.8) | 73 (7.7) | |
| **Place of delivery of the index child** | | | <0.001 |
| Health facility | 574 (60.5) | 218 (23) | |
| Home | 71 (7.5) | 86 (9) | |
| **Birth order of the index child** | | | 0.113 |
| First | 172 (18.1) | 83 (8.7) | |
| Second to fifth | 426 (44.9) | 187 (19.7) | |
| Sixth and above | 47 (5.0) | 34 (3.6) | |
| **Birth interval between older and index child (months)** | | | 0.414 |
| <36 | 84 (12.1) | 45 (6.5) | |
| ≥36 | 389 (56.1) | 176 (25.3) | |
| **PNC follow-up** | | | <0.001 |
| Yes | 328 (34.6) | 102 (10.7) | |
| No | 317 (33.4) | 202 (21.3) | |
| **Number of PNC follow-ups (n = 430)** | | | 0.130 |
| <3 | 292 (67.9) | 96 (22.3) | |
| ≥3 | 36 (8.4) | 6 (1.4) | |
| **Counseling on child feeding in ANC/PNC** | | | <0.001 |
| Yes | 371 (39.1) | 111 (11.7) | |
| No | 274 (28.9) | 193 (20.3) | |
| **GMP follow-up** | | | <0.001 |
| Yes | 449 (47.3) | 134 (14.1) | |
| No | 196 (20.7) | 170 (17.9) | |
| **Counseling about child feeding in GMP** | | | 0.011 |
| Yes | 411 (70.5) | 112 (19.2) | |
| No | 38 (6.5) | 22 (3.8) | |
| **HEW home visit in the previous month** | | | <0.001 |
| Yes | 323 (34) | 111 (11.7) | |
| No | 322 (33.9) | 193 (20.4) | |

* Chi-square or t-test.

In summary, in the study area, efforts are being made to prevent malnutrition through nutrition education, micronutrient supplementation, and deworming. The National Nutrition Program II (NNP II) and the Ethiopian Food and Nutrition Policy are also in place to implement nutrition-sensitive and nutrition-specific interventions through a coordinated multi-sectoral approach [44,45]. Regarding treatment, uncomplicated severe acute malnutrition (SAM) is treated in the outpatient

**Table 3. Factors associated with meal frequency among children aged 6–23 months.**

| Variable | COR (95% CI) | P-value | AOR (95% CI) | P-value |
|---|---|---|---|---|
| **Wealth index** | | | | |
| Poor | Reference | | | |
| Medium | 1.406 (1.018, 1.943) | 0.039 | 1.241 (.839, 1.836) | 0.280 |
| Rich | 2.270 (1.604, 3.213) | <0.001 | 1.701 (1.120, 2.585) | 0.013 |
| **GMP follow-up** | | | | |
| Yes | 2.906 (2.193, 3.852) | <0.001 | 2.034 (1.439, 2.875) | <0.001 |
| No | Reference | | | |
| **Mother's knowledge of child feeding** | | | | |
| Good | 3.602 (2.707, 4.792) | <0.001 | 2.261 (1.588, 3.219) | <0.001 |
| Poor | Reference | | | |
| **Birth preparedness plan** | | | | |
| Yes | 3.390 (2.315, 4.965) | <0.001 | 2.097 (1.013, 4.339) | 0.046 |
| No | Reference | | | |
| **Maternal social capital** | | | | |
| Good | 2.340 (1.765, 3.103) | <0.001 | 1.391 (0.975, 1.985) | 0.069 |
| Poor | Reference | | | |

therapeutic program (OTP). Meanwhile, complicated SAM is managed at the stabilization center (SC) as an inpatient treatment. Conversely, moderate acute malnutrition (MAM) is treated through the targeted supplementary feeding program (TSFP) [46]. Additionally, food security and feeding programs in the area include the Productive Safety Net Program (PSNP), emergency food assistance for vulnerable households, and the general food distribution (GFD). Despite all these efforts, malnutrition is still high [14,47], and inadequate meal frequency among young children can contribute to it, as shown in this study.

This study was conducted at the community level, which strengthens its generalizability to the source population, but its geographically limited scope restricts its overall generalizability. Moreover, the cross-sectional study design can show us associations, but not establish causality due to temporal ambiguity. The 24-hour recall method used to assess meal frequency may also have introduced recall and social desirability biases from respondents. In addition, the day-to-day meal frequency variations may not be adequately captured by a single 24-hour recall.

## Conclusions

The proportion of children who did not receive adequate meal frequency was significantly high. Being from a wealthy household, undergoing growth monitoring follow-up, having a knowledgeable mother regarding child feeding, and having a birth preparedness plan were associated with adequate meal frequency. In conclusion, our findings highlight the need to improve child meal frequency through enhancing maternal knowledge of child feeding, strengthening growth monitoring and promotion services, and improving socioeconomic status as indicated by wealth index.

## Supporting information

**S1 Dataset. Dataset used for the analysis. It contains all variables included in the study, and the data are anonymized.**
(DTA)

## Acknowledgments

We want to acknowledge Aksum University for supporting this study. We are also grateful to the study participants, data collectors, and supervisors for their participation in this study.

## Author contributions

**Conceptualization:** Shewit Engdashet Berhe, Ermyas Brhane.

**Data curation:** Shewit Engdashet Berhe.

**Formal analysis:** Shewit Engdashet Berhe, Teklit Grum, Teklehaymanot Huluf Abraha, Gebrekiros Aregawi, Kiros Gereziher Arefayne, Ermyas Brhane.

**Investigation:** Shewit Engdashet Berhe, Teklit Grum, Teklehaymanot Huluf Abraha, Gebrekiros Aregawi, Ebud Ayele Dagnazgi, Kiros Gereziher Arefayne, Ermyas Brhane.

**Methodology:** Shewit Engdashet Berhe, Teklit Grum, Teklehaymanot Huluf Abraha, Gebrekiros Aregawi, Ebud Ayele Dagnazgi, Kiros Gereziher Arefayne, Ermyas Brhane.

**Project administration:** Ermyas Brhane.

**Software:** Shewit Engdashet Berhe, Ebud Ayele Dagnazgi, Ermyas Brhane.

**Supervision:** Shewit Engdashet Berhe, Ermyas Brhane.

**Visualization:** Ermyas Brhane.

**Writing – original draft:** Shewit Engdashet Berhe, Ermyas Brhane.

**Writing – review & editing:** Shewit Engdashet Berhe, Ermyas Brhane.

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
