## [Decision Letter · Decision Letter 0]

26 Aug 2024

PGPH-D-24-01350

Meal frequency and its associated factors among children aged 6-23 months in the Tahtay Maichew district, northern Ethiopia

Dear Dr. Engdashet,

Thank you for submitting your manuscript to PLOS Global Public Health. After careful consideration, we feel that it has merit but does not fully meet PLOS Global Public Health’s publication criteria as it currently stands. Therefore, we invite you to submit a revised version of the manuscript that addresses the points raised during the review process.

We look forward to receiving your revised manuscript.

Kind regards,

Dessalegn Tamiru

Academic Editor

Journal Requirements:

Additional Editor Comments (if provided):

Reviewer 1

The study is significant, but the authors should pay attention to the concerns.

Abstract

Specify types of multivariable analysis.

How was the was the endpoint variable assessed?

Results

Specify the wealth index.

Conclusion

Expunge the word "recommendation"

Line 31-33 Statements were similar to the ones in the results.

Main body

Methods

The methodology has major defects.

Source population, study population, inclusion,inclusion and exclusion criteria were missed.

What is the justification for the 4% margin error? DE?

How many HH assets were assessed?

How did you include HH assets for further analysis (PCA)?

Mention the response variable before data processing and analysis.

What was the maximum value of VIF?

What was the justification for a p value < 0.1 for candidate predictors in the final model?

Why did you use mean and SD for age?

Ethical issues

Confidentiality issues were not mentioned.

Results

Do not interpret the findings.

Discussion

The discussion is very clumsy.

English writing and editorial issues are not optimal.

Reviewer 2

Meal frequency and its associated factors among children aged 6-23 months in the Tahtay Maichew district, northern Ethiopia

First of all, I appreciate the work you did but I have the following main questions

Why you are targeting infants or children aged 6-23 months?

What is the novelty of your study since many studies done with similar topics in the study area and country?

In general, the methods part needs major revision?

1. Abstract

The background information ¨ The term “meal frequency” describes the adequacy of complementary feedings among children¨ it would be better if you focused on or included the severity or magnitudes of the problems or the Gap.

Results: is not written clearly, so would you re-write again

Methods: We conducted a community-based cross-sectional study among 981 mothers with

index children aged 6-23 months. It would be better if changed to A community-based cross-sectional study was conducted among 981 mothers with index children aged 6-23 months

Conclusion and recommendation: would you write it based on your findings

2. Introduction: Para 2 and 3 are redundant as they have no direct relevance to the current objectives of the study, and are not required for the audience of journals dealing with PLOS Global Public Health. The introduction section is too short and not clear, and it still is unable to appropriately justify neither the need for this study nor the target age group.

3. Methods

The recruitment period for this study was from 22/1/2018 to 30/2/2018. How is it possible to collect such much data within specified data, n=981? I don’t think so? would you check it again?

Sample size determination

Why 4% margin of error, a nonresponse rate 10%. Why not 5% and NRR=5%?

Initially, out of the seventeen kebeles in the Tahtay Maichew district, eight kebeles were chosen using the lottery method. How did you select eight of them?

Did you any sampling techniques or randomization? if how?

What is your sampling frame?

What does mean Kebelle ?

Did you use PPS? how?

using a systematic random sampling technique versus a sampling frame? So how did you see those two things?

Your sampling procedure is not clear, please write it clearly.

Would you include operational definitions like?

PCA?

Knowledge good or poor?

Maternal Social capital good or poor?

Received minimum meal frequency?

Etc

What was your outcome variables?

In data management and analysis

Did you check the normality distribution of your data? how?

Write what you did for categorical and continuous variables.

For example

The mean age [±SD] of the children was 13.7 ±4.59 months.

The mean age [±SD] of the mothers was 29.74 ±6.66 years.

The mean family size [±SD] was 4.75 ±1.7 persons.

Not nothing mentioned in the methods part

4. Results

Table 1: Socio-demographic and socio-economic characteristics of participants

Do cross tab for Tables 1 and 2

Mothers age in years change it to age of mother (yrs)

Age of index child in months change it to Age of child (mo)

Family size in persons change it to family size

Etc , do the other like this

Do you think the Husband's educational status is important?

Did you assess GMP?

In the bivariate analysis

According to the bivariable logistic regression analysis, eight variables, namely, wealth index,

ANC follow-up, place of delivery, growth monitoring follow-up, health extension worker home visit, mother’s knowledge of child feeding, media exposure, and mother’s social capital in the last 12 months, were significantly associated with meal frequency at 95% CL (p ≤ 0.1).

why the reason you only select those variables p≤0.1, and why not p≤0.25?

5. Discussion

The discussion needs to be focused towards the main objectives. The functional significance of the present results may be elaborated.

Please would include the strengths and limitations of the study

What is the clinical implication of your findings?

6. Conclusion

Would you rewrite it based on your findings?

7. References

References need to be written in the style recommended by the journal PLOS Global Public Health.

For example, if you take reference numbers 1 and 2 are written wrongly

1. Organization, W.H., Infant and young child feeding: model chapter for textbooks for medical

students and allied health professionals. 2009: World Health Organization. Available from: include it

2. Organization, W.H., Indicators for assessing infant and young child feeding practices: part 1:

definitions: conclusions of a consensus meeting held 6-8 November 2007 in Washington DC, USA. 2008: World Health Organization . Available from:include it

Reviewers' comments:

Reviewer's Responses to Questions

**Comments to the Author**

1. Does this manuscript meet PLOS Global Public Health’s publication criteria ? Is the manuscript technically sound, and do the data support the conclusions? The manuscript must describe methodologically and ethically rigorous research with conclusions that are appropriately drawn based on the data presented.

Reviewer #1: Yes

Reviewer #2: Yes

2. Has the statistical analysis been performed appropriately and rigorously?

Reviewer #1: Yes

Reviewer #2: Yes

3. Have the authors made all data underlying the findings in their manuscript fully available (please refer to the Data Availability Statement at the start of the manuscript PDF file)?

Reviewer #1: No

Reviewer #2: No

4. Is the manuscript presented in an intelligible fashion and written in standard English?

Reviewer #1: No

Reviewer #2: No

5. Review Comments to the Author

Reviewer #1: The study is significant, but the authors should pay attention to the concerns.

Abstract

Specify types of multivariable analysis.

How was the was the endpoint variable assessed?

Results

Specify the wealth index.

Conclusion

Expunge the word "recommendation"

Line 31-33 Statements were similar to the ones in the results.

Main body

Methods

The methodology has major defects.

Source population, study population, inclusion,inclusion and exclusion criteria were missed.

What is the justification for the 4% margin error? DE?

How many HH assets were assessed?

How did you include HH assets for further analysis (PCA)?

Mention the response variable before data processing and analysis.

What was the maximum value of VIF?

What was the justification for a p value < 0.1 for candidate predictors in the final model?

Why did you use mean and SD for age?

Ethical issues

Confidentiality issues were not mentioned.

Results

Do not interpret the findings.

Discussion

The discussion is very clumsy.

English writing and editorial issues are not optimal.

Reviewer #2: Meal frequency and its associated factors among children aged 6-23 months in the Tahtay Maichew district, northern Ethiopia

First of all, I appreciate the work you did but I have the following main questions

Why you are targeting infants or children aged 6-23 months?

What is the novelty of your study since many studies done with similar topics in the study area and country?

In general, the methods part needs major revision?

1. Abstract

The background information ¨ The term “meal frequency” describes the adequacy of complementary feedings among children¨ it would be better if you focused on or included the severity or magnitudes of the problems or the Gap.

Results: is not written clearly, so would you re-write again

Methods: We conducted a community-based cross-sectional study among 981 mothers with

index children aged 6-23 months. It would be better if changed to A community-based cross-sectional study was conducted among 981 mothers with index children aged 6-23 months

Conclusion and recommendation: would you write it based on your findings

2. Introduction: Para 2 and 3 are redundant as they have no direct relevance to the current objectives of the study, and are not required for the audience of journals dealing with PLOS Global Public Health. The introduction section is too short and not clear, and it still is unable to appropriately justify neither the need for this study nor the target age group.

3. Methods

The recruitment period for this study was from 22/1/2018 to 30/2/2018. How is it possible to collect such much data within specified data, n=981? I don’t think so? would you check it again?

Sample size determination

Why 4% margin of error, a nonresponse rate 10%. Why not 5% and NRR=5%?

Initially, out of the seventeen kebeles in the Tahtay Maichew district, eight kebeles were chosen using the lottery method. How did you select eight of them?

Did you any sampling techniques or randomization? if how?

What is your sampling frame?

What does mean Kebelle ?

Did you use PPS? how?

using a systematic random sampling technique versus a sampling frame? So how did you see those two things?

Your sampling procedure is not clear, please write it clearly.

Would you include operational definitions like?

PCA?

Knowledge good or poor?

Maternal Social capital good or poor?

Received minimum meal frequency?

Etc

What was your outcome variables?

In data management and analysis

Did you check the normality distribution of your data? how?

Write what you did for categorical and continuous variables.

For example

The mean age [±SD] of the children was 13.7 ±4.59 months.

The mean age [±SD] of the mothers was 29.74 ±6.66 years.

The mean family size [±SD] was 4.75 ±1.7 persons.

Not nothing mentioned in the methods part

4. Results

Table 1: Socio-demographic and socio-economic characteristics of participants

Do cross tab for Tables 1 and 2

Mothers age in years change it to age of mother (yrs)

Age of index child in months change it to Age of child (mo)

Family size in persons change it to family size

Etc , do the other like this

Do you think the Husband's educational status is important?

Did you assess GMP?

In the bivariate analysis

According to the bivariable logistic regression analysis, eight variables, namely, wealth index,

ANC follow-up, place of delivery, growth monitoring follow-up, health extension worker home visit, mother’s knowledge of child feeding, media exposure, and mother’s social capital in the last 12 months, were significantly associated with meal frequency at 95% CL (p ≤ 0.1).

why the reason you only select those variables p≤0.1, and why not p≤0.25?

5. Discussion

The discussion needs to be focused towards the main objectives. The functional significance of the present results may be elaborated.

Please would include the strengths and limitations of the study

What is the clinical implication of your findings?

6. Conclusion

Would you rewrite it based on your findings?

7. References

References need to be written in the style recommended by the journal PLOS Global Public Health.

For example, if you take reference numbers 1 and 2 are written wrongly

1. Organization, W.H., Infant and young child feeding: model chapter for textbooks for medical

students and allied health professionals. 2009: World Health Organization. Available from: include it

2. Organization, W.H., Indicators for assessing infant and young child feeding practices: part 1:

definitions: conclusions of a consensus meeting held 6-8 November 2007 in Washington DC, USA. 2008: World Health Organization . Available from:include it

etc

6. PLOS authors have the option to publish the peer review history of their article (what does this mean? ). If published, this will include your full peer review and any attached files.

**Do you want your identity to be public for this peer review?** For information about this choice, including consent withdrawal, please see our Privacy Policy .

Reviewer #1: No

Reviewer #2: **Yes:** Melese Sinaga

---

## [Decision Letter · Decision Letter 1]

6 Jan 2025

PGPH-D-24-01350R1

Meal frequency and its associated factors among children aged 6-23 months in the Tahtay Maichew district, northern Ethiopia

Dear Dr. Engdashet,

Thank you for submitting your manuscript to PLOS Global Public Health. After careful consideration, we feel that it has merit but does not fully meet PLOS Global Public Health’s publication criteria as it currently stands. Therefore, we invite you to submit a revised version of the manuscript that addresses the points raised during the review process.

We look forward to receiving your revised manuscript.

Kind regards,

Dessalegn Tamiru

Academic Editor

Additional Editor Comments (if provided):

Reviewer 1

Reviewer Recommendation Term: Major Revision

Rate Review: 0

Custom Review Question(s): Response

Comments to the Author

1. Does this manuscript meet PLOS Global Public Health’s publication criteria? Is the manuscript technically sound, and do the data support the conclusions? The manuscript must describe methodologically and ethically rigorous research with conclusions that are appropriately drawn based on the data presented. Yes

2. Has the statistical analysis been performed appropriately and rigorously? Yes

3. Have the authors made all data underlying the findings in their manuscript fully available (please refer to the Data Availability Statement at the start of the manuscript PDF file)?

The PLOS Data policy requires authors to make all data underlying the findings described in their manuscript fully available without restriction, with rare exception. The data should be provided as part of the manuscript or its supporting information, or deposited to a public repository. For example, in addition to summary statistics, the data points behind means, medians and variance measures should be available. If there are restrictions on publicly sharing data—e.g. participant privacy or use of data from a third party—those must be specified. No

4. Is the manuscript presented in an intelligible fashion and written in standard English?

PLOS Global Public Health does not copyedit accepted manuscripts, so the language in submitted articles must be clear, correct, and unambiguous. Any typographical or grammatical errors should be corrected at revision, so please note any specific errors here. No

5. Review Comments to the Author

Please use the space provided to explain your answers to the questions above. You may also include additional comments for the author, including concerns about dual publication, research ethics, or publication ethics. (Please upload your review as an attachment if it exceeds 20,000 characters) The study is significant, but the authors should pay attention to the concerns.

Abstract

Specify types of multivariable analysis.

How was the was the endpoint variable assessed?

Results

Specify the wealth index.

Conclusion

Expunge the word "recommendation"

Line 31-33 Statements were similar to the ones in the results.

Main body

Methods

The methodology has major defects.

Source population, study population, inclusion,inclusion and exclusion criteria were missed.

What is the justification for the 4% margin error? DE?

How many HH assets were assessed?

How did you include HH assets for further analysis (PCA)?

Mention the response variable before data processing and analysis.

What was the maximum value of VIF?

What was the justification for a p value < 0.1 for candidate predictors in the final model?

Why did you use mean and SD for age?

Ethical issues

Confidentiality issues were not mentioned.

Results

Do not interpret the findings.

Discussion

The discussion is very clumsy.

English writing and editorial issues are not optimal.

Reviewer 2

Reviewer Recommendation Term: Major Revision

Rate Review: 0

Custom Review Question(s): Response

Comments to the Author

1. Does this manuscript meet PLOS Global Public Health’s publication criteria? Is the manuscript technically sound, and do the data support the conclusions? The manuscript must describe methodologically and ethically rigorous research with conclusions that are appropriately drawn based on the data presented. Yes

2. Has the statistical analysis been performed appropriately and rigorously? Yes

3. Have the authors made all data underlying the findings in their manuscript fully available (please refer to the Data Availability Statement at the start of the manuscript PDF file)?

The PLOS Data policy requires authors to make all data underlying the findings described in their manuscript fully available without restriction, with rare exception. The data should be provided as part of the manuscript or its supporting information, or deposited to a public repository. For example, in addition to summary statistics, the data points behind means, medians and variance measures should be available. If there are restrictions on publicly sharing data—e.g. participant privacy or use of data from a third party—those must be specified. No

4. Is the manuscript presented in an intelligible fashion and written in standard English?

PLOS Global Public Health does not copyedit accepted manuscripts, so the language in submitted articles must be clear, correct, and unambiguous. Any typographical or grammatical errors should be corrected at revision, so please note any specific errors here. No

5. Review Comments to the Author

Please use the space provided to explain your answers to the questions above. You may also include additional comments for the author, including concerns about dual publication, research ethics, or publication ethics. (Please upload your review as an attachment if it exceeds 20,000 characters) Meal frequency and its associated factors among children aged 6-23 months in the Tahtay Maichew district, northern Ethiopia

First of all, I appreciate the work you did but I have the following main questions

Why you are targeting infants or children aged 6-23 months?

What is the novelty of your study since many studies done with similar topics in the study area and country?

In general, the methods part needs major revision?

1. Abstract

The background information ¨ The term “meal frequency” describes the adequacy of complementary feedings among children¨ it would be better if you focused on or included the severity or magnitudes of the problems or the Gap.

Results: is not written clearly, so would you re-write again

Methods: We conducted a community-based cross-sectional study among 981 mothers with

index children aged 6-23 months. It would be better if changed to A community-based cross-sectional study was conducted among 981 mothers with index children aged 6-23 months

Conclusion and recommendation: would you write it based on your findings

2. Introduction: Para 2 and 3 are redundant as they have no direct relevance to the current objectives of the study, and are not required for the audience of journals dealing with PLOS Global Public Health. The introduction section is too short and not clear, and it still is unable to appropriately justify neither the need for this study nor the target age group.

3. Methods

The recruitment period for this study was from 22/1/2018 to 30/2/2018. How is it possible to collect such much data within specified data, n=981? I don’t think so? would you check it again?

Sample size determination

Why 4% margin of error, a nonresponse rate 10%. Why not 5% and NRR=5%?

Initially, out of the seventeen kebeles in the Tahtay Maichew district, eight kebeles were chosen using the lottery method. How did you select eight of them?

Did you any sampling techniques or randomization? if how?

What is your sampling frame?

What does mean Kebelle ?

Did you use PPS? how?

using a systematic random sampling technique versus a sampling frame? So how did you see those two things?

Your sampling procedure is not clear, please write it clearly.

Would you include operational definitions like?

PCA?

Knowledge good or poor?

Maternal Social capital good or poor?

Received minimum meal frequency?

Etc

What was your outcome variables?

In data management and analysis

Did you check the normality distribution of your data? how?

Write what you did for categorical and continuous variables.

For example

The mean age [±SD] of the children was 13.7 ±4.59 months.

The mean age [±SD] of the mothers was 29.74 ±6.66 years.

The mean family size [±SD] was 4.75 ±1.7 persons.

Not nothing mentioned in the methods part

4. Results

Table 1: Socio-demographic and socio-economic characteristics of participants

Do cross tab for Tables 1 and 2

Mothers age in years change it to age of mother (yrs)

Age of index child in months change it to Age of child (mo)

Family size in persons change it to family size

Etc , do the other like this

Do you think the Husband's educational status is important?

Did you assess GMP?

In the bivariate analysis

According to the bivariable logistic regression analysis, eight variables, namely, wealth index,

ANC follow-up, place of delivery, growth monitoring follow-up, health extension worker home visit, mother’s knowledge of child feeding, media exposure, and mother’s social capital in the last 12 months, were significantly associated with meal frequency at 95% CL (p ≤ 0.1).

why the reason you only select those variables p≤0.1, and why not p≤0.25?

5. Discussion

The discussion needs to be focused towards the main objectives. The functional significance of the present results may be elaborated.

Please would include the strengths and limitations of the study

What is the clinical implication of your findings?

6. Conclusion

Would you rewrite it based on your findings?

7. References

References need to be written in the style recommended by the journal PLOS Global Public Health.

For example, if you take reference numbers 1 and 2 are written wrongly

1. Organization, W.H., Infant and young child feeding: model chapter for textbooks for medical

students and allied health professionals. 2009: World Health Organization. Available from: include it

2. Organization, W.H., Indicators for assessing infant and young child feeding practices: part 1:

definitions: conclusions of a consensus meeting held 6-8 November 2007 in Washington DC, USA. 2008: World Health Organization . Available from:include it

etc

Reviewers' comments:

Reviewer's Responses to Questions

**Comments to the Author**

1. If the authors have adequately addressed your comments raised in a previous round of review and you feel that this manuscript is now acceptable for publication, you may indicate that here to bypass the “Comments to the Author” section, enter your conflict of interest statement in the “Confidential to Editor” section, and submit your "Accept" recommendation.

Reviewer #1: (No Response)

2. Does this manuscript meet PLOS Global Public Health’s publication criteria ? Is the manuscript technically sound, and do the data support the conclusions? The manuscript must describe methodologically and ethically rigorous research with conclusions that are appropriately drawn based on the data presented.

Reviewer #1: No

3. Has the statistical analysis been performed appropriately and rigorously?

Reviewer #1: No

4. Have the authors made all data underlying the findings in their manuscript fully available (please refer to the Data Availability Statement at the start of the manuscript PDF file)?

Reviewer #1: No

5. Is the manuscript presented in an intelligible fashion and written in standard English?

Reviewer #1: Yes

6. Review Comments to the Author

Reviewer #1: Meal frequency and its associated factors among children aged 6-23 months in the Tahtay Maichew district, northern Ethiopia.

The authors have attempted to respond to the concerns; however, it is not fully addressed as requested.

Abstract

Line 14: Replace introduction by background.

Line 19: Indicate the research gap rather than stating ‘little is known about meal frequency...

Specify outcome assessment in the methods

Introduction

Line 71: indicate the research gap clearly.

Methods

There are still methodological flaws.

Lines 90-99: Restate it in paragraph form.

The authors did not specify the threshold for how HH assets were selected for further analysis.

The authors are recommended to redo analysis by taking a threshold, a p value < 0.05, as the authors did not present scientifically convincing argumentative evidence.

The limitations of the report should be fully explained, and overstated conclusions are expected to be revised.

7. PLOS authors have the option to publish the peer review history of their article (what does this mean? ). If published, this will include your full peer review and any attached files.

**Do you want your identity to be public for this peer review?** For information about this choice, including consent withdrawal, please see our Privacy Policy .

Reviewer #1: No

---

## [Editor Report · Decision Letter 2]

27 Jun 2025

PGPH-D-24-01350R2

Meal frequency and its associated factors among children aged 6-23 months in the Tahtay Maichew district, northern Ethiopia

Dear Dr. Engdashet,

Thank you for submitting your manuscript to PLOS Global Public Health. After careful consideration, we feel that it has merit but does not fully meet PLOS Global Public Health’s publication criteria as it currently stands. Therefore, we invite you to submit a revised version of the manuscript that addresses the points raised during the review process.

We look forward to receiving your revised manuscript.

Kind regards,

Nancy Angeline Gnanaselvam

Academic Editor

Additional Editor Comments:

The study has been done meticulously in field but needs major revision in the write up of manuscript

Title: Meal frequency or dietary assessment?

Study participants: Why mothers and not parent?

Elaborate on poverty level, any conflicts in study area

Key words: Select up to 10 key words from MESH

Line no 54-56: If things go wrong during this time we only have little opportunity to correct them later and the consequences may be devastating concerning the health and future of the child - Rewrite in professional English

Introduction: Mention about exclusive breastfeeding and continued breastfeeding rates in the study area

Line no 83: Duration of study mention only year and not date

The weather conditions of the district are semi-highland: Check the necessity of this statement

Line no 89: Populations or study participants?

Add ethical approval statement in methods in the beginning itself

International Code of Marketing of Breast-milk Substitutes : Adherence to the code in the study area needs to be mentioned in methods or introduction

Elaborate operational definitions and variables description is not required - Please rewrite concisely

The results need more elaboration on Breastfeeding, types of foods consumed- with break up of macro and micronutrients, consumption of junk food, screen time, anthropometric status of children, any illness

Presentation of results in very basic, with less efforts from authors to regroup variables and make the tables in presentable fashion

Tables and presentation of data can be improved by making it concise

Wherever you are comparing meal frequency add a p value and statistic. mention details in legend

What is the MAM and SAM treatment protocol in the study area - are there any food security, feeding programs in the study area

What does the health system in study area do to ensure regular growth monitoring? What are the measures to ensure universal health coverage of malnutrition prevention in the study area?

What are the existing health system interventions to prevent malnutrition in the study area?

Discussion is not written with the public health angle of the problem studied - it needs to be rewritten using the suggestions given by me

You can elaborate on best practices from other resource limited settings in which dietary intake was ensured among under 5 children
---

## [Editor Report · Decision Letter 3]

17 Sep 2025

PGPH-D-24-01350R3

Feeding pattern and associated factors among children aged 6-23 months in the Tahtay Maichew district, northern Ethiopia

Dear Dr. Engdashet,

Thank you for submitting your manuscript to PLOS Global Public Health. After careful consideration, we feel that it has merit but does not fully meet PLOS Global Public Health’s publication criteria as it currently stands. Therefore, we invite you to submit a revised version of the manuscript that addresses the points raised during the review process.

We look forward to receiving your revised manuscript.

Kind regards,

Nancy Angeline Gnanaselvam

Academic Editor

Journal Requirements:

Additional Editor Comments (if provided):

While there was response to the reviewer's comments, Authors have attempted only superficially attempted to address the reviewers' concerns. Substantial revision of the paper has not been undertaken. There was Northern Ethiopian Conflict which could or not have impacted the food security. Public health aspects such as health system, female literacy, age of marriage, spacing, family size etc needs to be focussed. MDD-IYCF as per UNICEF could have been measured but it is not focssed in the paper. Kindly rewrite the manuscript, strengthen introduction, results and discussion and submit. Key words need to be upto 10

Reviewers' comments:

Figure Resubmissions:

---

## [Editor Report · Decision Letter 4]

27 Nov 2025

PGPH-D-24-01350R4

Feeding pattern and associated factors among children aged 6-23 months in the Tahtay Maichew district, northern Ethiopia

Dear Dr. Engdashet,

Thank you for submitting your manuscript to PLOS Global Public Health. After careful consideration, we feel that it has merit but does not fully meet PLOS Global Public Health’s publication criteria as it currently stands. Therefore, we invite you to submit a revised version of the manuscript that addresses the points raised during the review process.

We look forward to receiving your revised manuscript.

Kind regards,

Nancy Angeline Gnanaselvam

Academic Editor

Journal Requirements:

Additional Editor Comments (if provided):

Abstract: Grammatical errors need to be corrected. Mention study site

Key words: Add MESH terms such as Malnutrition, stunting etc

Study setting section needs to be added in methods

Discussion can be strengthened by adding details about food security of the region

Conflict of interest and author contributions section need to be mentioned

Reviewers' comments:

 Figure Resubmissions:

---

## [Editor Report · Decision Letter 5]

8 Jan 2026

Feeding pattern and associated factors among children aged 6-23 months in the Tahtay Maichew district, northern Ethiopia

PGPH-D-24-01350R5

Dear Mr. Engdashet,

We are pleased to inform you that your manuscript 'Feeding pattern and associated factors among children aged 6-23 months in the Tahtay Maichew district, northern Ethiopia' has been provisionally accepted for publication in PLOS Global Public Health.

Best regards,

Nancy Angeline Gnanaselvam

Academic Editor
